# Endobronchial Ultrasound Using Guide Sheath-Guided Transbronchial Lung Biopsy in Ground-Glass Opacity Pulmonary Lesions without Fluoroscopic Guidance

**DOI:** 10.3390/cancers16061203

**Published:** 2024-03-19

**Authors:** Jongsoo Park, Changwoon Kim, Jong Geol Jang, Seok Soo Lee, Kyung Soo Hong, June Hong Ahn

**Affiliations:** 1Department of Radiology, Yeungnam University Medical Center, College of Medicine, Yeungnam University, Daegu 42415, Republic of Korea; jongsoo8393@naver.com; 2Division of Pulmonology, Department of Internal Medicine, College of Medicine, Yeungnam University, Daegu 42415, Republic of Korea; imkcw1127@gmail.com (C.K.); jonggirl83@naver.com (J.G.J.); 3Department of Thoracic and Cardiovascular Surgery, College of Medicine, Yeungnam University, Daegu 42415, Republic of Korea; andrea0710@naver.com

**Keywords:** lung cancer, ground glass opacity, bronchoscopy

## Abstract

**Simple Summary:**

Lung cancer screening programs have led to the increased detection of pulmonary ground-glass opaque lesions. An accurate diagnosis of these lesions is essential in clinical practice. This study assessed the effectiveness of radial probe endobronchial ultrasound-guided transbronchial lung biopsy (RP-EBUS-TBLB) in diagnosing ground-glass opacity (GGO) pulmonary lesions. Out of 1651 procedures, 115 GGO lesions were analyzed, with an 80.1% visualization yield and a 60% diagnostic success rate. Diagnosed lesions were larger (21.9 ± 7.3 mm) compared to undiagnosed ones (17.1 ± 6.6 mm, *p* < 0.001). Diagnostic yield varied by lesion size: 50% for <20 mm, 65.1% for 20–30 mm, and 85.7% for >30 mm lesions. The mixed blizzard sign on EBUS appeared in 60.6% of mixed GGO lesions, significantly associated with diagnostic success (OR, 20.92; CI, 7.50–58.31; *p* < 0.001). RP-EBUS-TBLB is a viable diagnostic approach for GGO pulmonary lesions with acceptable complications (1.7% pneumothorax, 2.6% hemoptysis).

**Abstract:**

Diagnosing ground-glass opacity (GGO) pulmonary lesions poses challenges. This study evaluates the utility of radial probe endobronchial ultrasound-guided transbronchial lung biopsy (RP-EBUS-TBLB) in diagnosing GGO pulmonary lesions. A total of 1651 RP-EBUS procedures were performed during the study period. This study analyzed 115 GGO lesions. The EBUS visualization yield was 80.1%. Of 115 lesions, 69 (60%) were successfully diagnosed. The average size of diagnosed lesions was significantly larger than that of undiagnosed lesions (21.9 ± 7.3 vs. 17.1 ± 6.6 mm, *p* < 0.001). Diagnostic yield varied by lesion size: 50.0% for lesions <20 mm, 65.1% for 20–30 mm lesions, and 85.7% for lesions >30 mm. The mixed blizzard sign on EBUS appeared in 60.6% of mixed GGO lesions, with no cases in pure GGO lesions. Multivariable analyses showed that lesion size (odds ratio [OR], 1.10; 95% confidence interval [CI], 1.00–1.16; *p* < 0.001) and mixed blizzard sign on EBUS (OR, 20.92; CI, 7.50–58.31; *p* < 0.001) were significantly associated with diagnostic success. Pneumothorax and hemoptysis occurred in 1.7% and 2.6% of patients, respectively. RP-EBUS-TBLB without fluoroscopic guidance is a viable diagnostic approach for GGO pulmonary lesions with acceptable complications.

## 1. Introduction

In recent years, the implementation of lung cancer screening programs in various countries has led to the increased detection of pulmonary ground-glass opacity (GGO) lesions [1,2]. This trend is particularly notable in Asian countries, where lung adenocarcinomas often present as GGOs, further amplified by the widespread use of computed tomography (CT) scans [3,4]. It is important to recognize that GGO pulmonary lesions are not exclusively indicative of cancer; they may also arise from various inflammatory conditions, localized fibrosis, organizing pneumonia, and other medical conditions [5]. An accurate diagnosis of these lesions is therefore essential in clinical practice.

The current guidelines from the Fleischner Society (2017) and Lung-RADS predominantly focus on follow-up strategies [6,7]. However, in practical clinical settings, lung cancers are frequently diagnosed at sizes smaller than those stipulated in these guidelines [8]. Recent studies have suggested that foregoing a histological diagnosis before surgery does not impact prognosis when the size of the solid component of a lesion increases during follow-up or is >5 mm [9,10]. Nevertheless, obtaining a preoperative histological diagnosis could potentially increase diagnostic precision.

While image-guided transthoracic needle biopsy has been effective for GGO lesions in previous studies [11,12], it faces challenges when lesions are deep in the lung or near blood vessels, raising concerns about complications and the risk of pleural seeding [12,13]. Surgical resection for both diagnosis and treatment is an alternative, but there is ongoing debate regarding the timing of surgery and the potential for resecting benign pathology [10].

Recent developments in radial probe endobronchial ultrasound-guided transbronchial lung biopsy (RP-EBUS-TBLB) have made it a promising modality for accurately diagnosing GGO lesions [14,15]. However, the diagnostic accuracy of and complication risks associated with RP-EBUS-TBLB for GGO lesions have not been comprehensively established. This study aims to assess both the diagnostic accuracy and associated complications of RP-EBUS-TBLB in the context of GGO lesions.

## 2. Materials and Methods

### 2.1. Study Design and Subjects

A total of 1651 RP-EBUS procedures without fluoroscopy were performed during the study period. This study involved a retrospective review of the medical records of 115 patients with GGO pulmonary lesions who underwent RP-EBUS-TBLB without fluoroscopic guidance at Yeungnam University Hospital from January 2019 to March 2022. The patient group consisted of 11 patients with pure GGO and 104 with mixed GGO. Histological examination was conducted unless the EBUS image was invisible (Figure 1). The study adhered to the principles of the Declaration of Helsinki and was approved by the institutional review board of Yeungnam University Hospital (YUH IRB 2020-09-025). The requirement for informed consent was waived due to the retrospective nature of the study.

### 2.2. CT and Bronchoscopy

Prior to the procedure, all patients underwent thin-section chest CT scans (0.75 mm slice thickness, 0.75 mm intervals; SOMATOM Definition AS 64-slice CT system; Siemens Healthcare, Erlangen, Germany). An experienced radiologist (J.S.P.) determined the CT characteristics of lung lesions (GGO or mixed GGO). The bronchus sign on CT was defined as a bronchus leading directly to the lesion. The shortest distance from the lesion to the pleura was measured on an axial plane CT scan, as previously described [16]. The EBUS image (blizzard, mixed blizzard, and invisible) was assessed by experienced respiratory physicians (K.S.H., J.G.J., and J.H.A.). Representative cases of RP-EBUS-TBLB for GGO pulmonary lesions are shown in Figure 2.

Regarding bronchoscopy, a 4 mm bronchoscope (BF P260F, Olympus, Tokyo, Japan) was used to reach the bronchus closest to the target lesion. Then, a RP-EBUS (UM S20–17S, Olympus) was inserted inside a GS through the bronchoscope working channel. Following the discovery of the GGO lesion, the RP was then removed, leaving the GS in place. Then, bronchial brush and biopsy forceps were introduced into the GS and brushings and biopsy specimens were collected. A GS kit (K-201; Olympus, Tokyo, Japan) including a forceps, and brush were used to biopsy the tissue. We obtained at least six tissues if there was no sign of complications.

### 2.3. Diagnostic Classification

Malignancy was diagnosed based on definite histological evidence. Benign lung lesions were diagnosed using criteria including the identification of definitive benign features and lesion regression with or without medical treatment. In this study, “diagnosed” cases were those conclusively identified as malignant or benign through RP-EBUS. Cases with indeterminate diagnoses (neither malignancy nor benign), and those where histological examination could not be performed due to EBUS image invisibility, were categorized as “undiagnosed”. The undiagnosed group was finally diagnosed using other tests such as antibiotic use, surgery, percutaneous needle biopsy (PCNB), and CT scan follow-up. If a GGO lesion persisted, they were followed up for at least 1 year, and a GGO lesion without change for 1 year was defined as stable disease. If a patient was lost during the CT follow-up period, we defined it as follow-up loss.

### 2.4. Statistical Analyses

Continuous variables were compared using Student’s *t*-test or the Mann–Whitney U test and are expressed as mean ± standard deviation. Categorical variables were analyzed using the chi-square test or Fisher’s exact test and are presented as frequencies (percentages). The EBUS visualization yield was calculated by dividing the number of cases with blizzard or mixed blizzard EBUS images by the total number of cases. The diagnostic yield was determined by dividing the number of diagnosed cases by the total number of cases. Factors influencing diagnostic success were identified through multivariable logistic regression analyses of factors with *p*-values < 0.1 in univariable analyses. In all analyses, *p* < 0.05 (two-tailed) was considered statistically significant. Statistical procedures were conducted using SPSS (version 24.0; IBM Corp., Armonk, NY, USA).

## 3. Results

### 3.1. Baseline Characteristics and Diagnostic Performance

The baseline characteristics of the 115 patients, categorized by diagnostic status, are detailed in Table 1. In the diagnosed group, the mean lesion size was notably larger (21.9 ± 7.3 vs. 17.1 ± 6.6 mm, *p* < 0.001). The presence of a positive CT bronchus sign was more common in the diagnosed group (79.7% vs. 39.1%, *p* < 0.001). Regarding procedure-related factors, the mixed blizzard EBUS image was more frequently observed in the diagnosed group. The procedure time was significantly shorter in the undiagnosed group (25.5 ± 11.1 vs. 21.2 ± 9.7 min, *p* = 0.036). Within the diagnosed group, adenocarcinoma was the most common diagnosis (88.4%), followed by organizing pneumonia (10.1%) and sarcoidosis (1.4%). Within the undiagnosed group, adenocarcinoma was the most common final diagnosis (52.2%), followed by stable disease (28.3%), follow-up loss (17.4%), and organizing pneumonia (4.2%). Regarding the final diagnosis method for the undiagnosed group, 24 adenocarcinoma cases were diagnosed by surgical resection (15 cases) and PCNB (9 cases). One organizing pneumonia case was diagnosed by the disappearance of the lesion on CT follow-up after antibiotics.

The diagnostic performance of RP-EBUS-TBLB in GGO pulmonary lesions is summarized in Figure 1. The EBUS visualization yield was 80.1% (93/115). Of the 115 lesions, 69 (60%) were successfully diagnosed by RP-EBUS-TBLB (4/11 pure GGO lesions and 65/104 mixed GGO lesions). The diagnostic yield of the EBUS guide sheath (GS) based on lesion size is presented in Table 2. The yield was 50.0% (29/58) for lesions <20 mm, 65.1% (28/43) for 20–30-mm lesions, and 85.7% (12/14) for lesions >30 mm.

### 3.2. Correlation between Chest CT Findings and EBUS Image

Table 3 shows the correlation between chest CT findings and EBUS image for GGO pulmonary lesions diagnosed by radial EBUS. Blizzard signs were found in 81.8% (9/11) of pure GGO lesions. Mixed blizzard signs were observed in 60.6% (63/104) of mixed GGO lesions. Notably, no mixed blizzard signs were found in pure GGO lesions.

### 3.3. Factors Affecting Diagnostic Success

Factors influencing diagnostic success are outlined in Table 4. Univariable analyses identified lesion size (odds ratio [OR], 1.11; 95% confidence interval [CI], 1.04–1.18; *p* = 0.001), positive bronchus sign (OR, 6.11; CI, 2.66–14.07; *p* < 0.001), and mixed blizzard sign on EBUS (OR, 24.00; CI, 8.78–65.61; *p* < 0.001) as significant contributors to diagnostic success. Multivariable analyses further indicated that lesion size (OR, 1.10; CI, 1.00–1.2; *p* = 0.042) and mixed blizzard sign on EBUS (OR, 20.92; CI, 7.50–58.31; *p* < 0.001) were associated with successful diagnosis.

### 3.4. Complications

In the cohort of 115 patients, the incidences of pneumothorax and hemoptysis were 1.7% (2/115) and 2.6% (3/115), respectively.

## 4. Discussion

In our study, we observed a diagnostic yield of 60% (n = 69/115) and a complication rate of 4% (n = 5/115), aligning with results reported in previous studies for GGO-predominant lesions [14,15,17]. Independent factors for diagnostic success were the mixed blizzard sign on EBUS (OR: 20.92; *p* < 0.001) and the size of the lesion (OR: 1.10; *p* = 0.042). Notably, the mixed blizzard sign on EBUS was present in 60.6% (63/104) of mixed GGO lesions but was absent in pure GGO lesions, consistent with previous findings [17]. This study was the first to evaluate the efficacy and safety of RP-EBUS-TBLB in GGO pulmonary lesions without fluoroscopic guidance.

A preoperative biopsy of GGO lesions could theoretically prevent unnecessary surgery for benign pathology and reduce the time spent on operative diagnosis via frozen section biopsy in wedge resection. However, challenges in preoperative tissue biopsy, along with concerns about complications and false-negative results, are common in real clinical practice [13]. Percutaneous transthoracic needle biopsy (PTNB) is accurate and relatively safe for lung lesions, including GGO lesions, with an accuracy reported at ≥90% in previous studies [11,12,18,19]. However, complications are relatively frequent, ranging from 16 to 62% in previous meta-analyses [12,19]. Additionally, concerns remain about pleural recurrence, particularly in subpleural lesions [20,21,22].

Several studies have shown high diagnostic accuracy through direct surgery without preoperative tissue biopsy, given the risks of side effects, possible underestimation of diagnosis, false negatives, and concerns about pleural recurrence with PTNB [10,23,24]. Cho et al. reported high diagnostic yield with surgical resection without preoperative tissue biopsy, especially in carefully selected GGO lesions highly suspicious for malignancy. This approach also led to reduced costs and shorter hospital stays, though ~5% of surgeries were unnecessary [23]. However, intraoperative tissue analysis based on frozen sections can extend surgical time and impact postoperative morbidity [25,26]. Therefore, if feasible, preoperative histological confirmation can improve diagnostic accuracy and reduce unnecessary surgeries and operative time.

In the current National Comprehensive Cancer Network (NCCN) Clinical Practice Guidelines in Oncology (NCCN Guidelines) for non-small cell lung cancer (version 2.2023) and lung cancer screening (version 1.2022), intraoperative or preoperative tissue biopsy is recommended for part-solid nodules with a solid component ≥8 mm. For pure GGO lesions, the guidelines suggest tissue biopsy for lesions ≥20 mm or those that have grown by ≥1.5 mm. However, the selection of biopsy method should be based on a multidisciplinary assessment of the difficulty and associated risks [27,28]. The proper use of PTNB, surgical resection, and bronchoscopy is vital, yet there are no definitive criteria, leading to ongoing debates regarding these methods. Although not explicitly mentioned in the guidelines, previous research highlights the benefits of radial EBUS for tissue biopsy of nodules.

Izumo et al. discussed the relationship between chest CT findings and EBUS images [17]. They noted that EBUS images are crucial for locating GGO pulmonary lesions. The blizzard sign, a whitish acoustic shadow observed while scanning from normal lung tissue to a GGO lesion, is noteworthy. In mixed GGO lesions, a mixed blizzard sign may indicate the preferred area for biopsy. In our study, the mixed blizzard sign was an independent factor for diagnostic success. Although the blizzard sign is a specific EBUS finding for GGO lesions, the detection of the mixed blizzard sign on EBUS is associated with a higher diagnosis rate for mixed GGO lesions, as observed in our study.

Ikezawa et al. initially reported a 57% (38/67) diagnostic success rate for peripheral GGO predominant-type lesions using TBLB with EBUS-GS [14]. They found that lesion size >25 mm and visibility under X-ray fluoroscopy were associated with diagnostic success. In their study, the diagnosis rate for total GGO lesions >25 mm was 94% (15/16). Ikezawa et al. later reported a 69% (156/169) diagnostic yield for 169 GGO pulmonary lesions using GS and virtual bronchoscopic navigation (VBN) [29], with a type 1 CT sign (a bronchus leading to the center of the lesion) as an independent factor for success. Our study, despite not using fluoroscopy, showed comparable diagnostic rates (60%, n = 69/115). Similar to previous studies, we observed a significant increase in the diagnosis rate with increasing lesion size, reaching 85.7% (n = 12/14) for lesions >30 mm.

This study was not without limitations. First, this was a retrospective study with a single-center setting and a relatively small number of GGO pulmonary lesions, which limits generalizability. Future studies should involve large, multicenter cohorts. Second, diagnostic yield may vary depending on the practitioner, but all operators in our study were trained in RP-EBUS-TBLB and followed the same protocol. The diagnostic yield could be influenced by additional modalities; our study used TBLB with GS without fluoroscopic guidance or VBN. In our center, we could not afford fluoroscopy and VBN, so we performed RP-EBUS-TBLB using only GS. The use of fluoroscopic guidance and VBN might improve diagnosis rates in some GGO pulmonary lesions. Finally, many lung cancers are associated with smoking and smoking status may affect diagnostic yield. This study is limited by not including smoking status as a factor. However, considering that most patients with GGO lesions are never smokers, the effect of smoking status in our study is unlikely to be significant. Further research is needed in this area. However, the strength of our study lies in diagnosing GGO pulmonary lesions without fluoroscopic guidance, achieving diagnostic rates comparable to previous studies.

## 5. Conclusions

RP-EBUS-TBLB using GS without fluoroscopy can be considered a viable diagnostic method for GGO pulmonary lesions. The EBUS image with the mixed blizzard sign and a large lesion were key factors for diagnostic success. The complication rates were within acceptable limits.

## Figures and Tables

**Figure 1 cancers-16-01203-f001:**
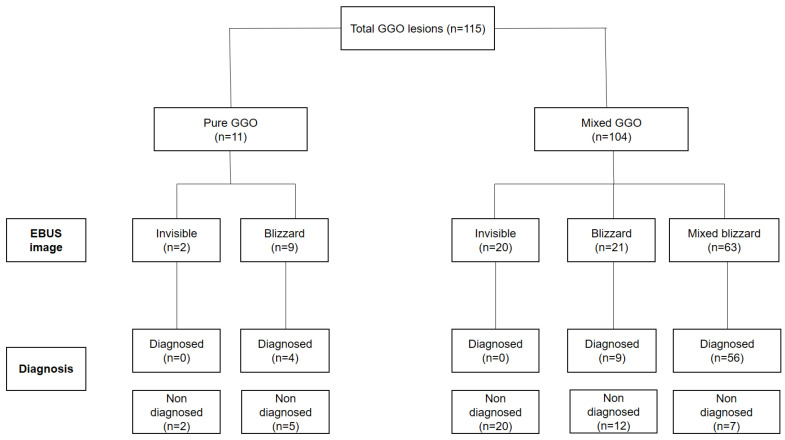
Study flowchart for 115 GGO pulmonary lesions. EBUS: endobronchial ultrasound; GGO: ground-glass opacity.

**Figure 2 cancers-16-01203-f002:**
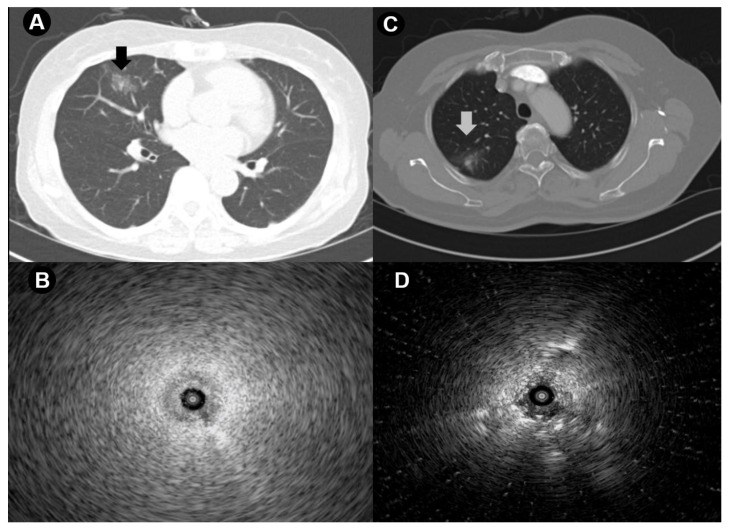
Representative cases. Schemes follow another format. If there are multiple panels, they should be listed as (**A**) a 68-year-old woman with pure GGO in the lateral segment of the right middle lobe in whom (**B**) EBUS revealed a blizzard sign, and the biopsy indicated focal atypical pneumocyte proliferation, suggestive of adenocarcinoma; (**C**) a 74-year-old woman with mixed GGO in the posterior segment of the right upper lobe in whom (**D**) EBUS demonstrated a mixed blizzard sign, and the biopsy confirmed adenocarcinoma.

**Table 1 cancers-16-01203-t001:** Baseline characteristics of GGO pulmonary lesions (n = 115).

Characteristic	Diagnosed (n = 69)	Undiagnosed (n = 46)	*p*-Value
Patients			
Age, years	65.7 ± 10.2	66.2 ± 10.7	0.830
Male sex	24 (34.8)	16 (34.8)	1.000
Location of lesions			0.126
Right upper lobe	21 (30.4)	18 (39.1)	
Right middle lobe	7 (10.1)	1 (2.2)	
Right lower lobe	12 (17.4)	9 (19.6)	
Left upper lobe	18 (26.1)	16 (34.8)	
Left lower lobe	11 (15.9)	2 (4.3)	
Characteristics			0.113
Pure GGO	4 (5.8)	7 (15.2)	
Mixed GGO	65 (94.2)	39 (84.8)	
Size (mm), long axis	21.9 ± 7.3	17.1 ± 6.6	<0.001
Distance from pleura (mm)	13.9 ± 11.1	15.5 ± 11.8	0.479
Bronchus sign in CT			<0.001
Positive	55 (79.7)	18 (39.1)	
Negative	14 (20.3)	28 (60.9)	
Procedure			
EBUS image			<0.001
Blizzard	13 (18.8)	17 (37.0)	
Mixed blizzard	56 (81.2)	7 (15.2)	
Invisible	0 (0.0)	22 (47.8)	
Procedure time, min	25.5 ± 11.1	21.2 ± 9.7	0.036
Complications			1.000
Pneumothorax	1 (1.4)	1 (2.2)	
Hemoptysis	2 (2.9)	1 (2.2)	
Diagnosis			
Adenocarcinoma	61 (88.4)	24 (52.2)	
Organizing pneumonia	7 (10.1)	1 (4.2)	
Sarcoidosis	1 (1.4)		
Stable disease		13 (28.3)	
Follow-up loss		8 (17.4)	

Values are presented as mean ± standard deviation or number (%). CT: computed tomography; EBUS: endobronchial ultrasound; GGO: ground-glass opacity.

**Table 2 cancers-16-01203-t002:** Diagnostic yield of EBUS-GS according to lesion size.

Lesion Size, mm	Total	Pure GGO	Mixed GGO
<20	29/58 (50.0)	3/7 (42.9)	26/51 (51.0)
20–30	28/43 (65.1)	0/3 (0.0)	28/40 (70.0)
>30	12/14 (85.7)	1/1 (80.0)	11/13 (84.6)
Total	69/115 (60.0)	4/11 (36.3)	65/104 (62.5)

Values are presented as number (%). EBUS: endobronchial ultrasound; GGO: ground-glass opacity; GS: guide sheath

**Table 3 cancers-16-01203-t003:** Correlation between chest CT findings and EBUS image.

	Pure GGO	Mixed GGO	*p*-Value
Subjects, n	11	104	
EBUS image			<0.001
Blizzard	9/11 (81.8)	21/104 (20.2)	
Mixed blizzard	0/11 (0)	63/104 (60.6)	
Invisible	2/11 (18.2)	20/104 (19.2)	

Values are presented as number (%). CT: computed tomography; EBUS: endobronchial ultrasound; GGO: ground-glass opacity

**Table 4 cancers-16-01203-t004:** Logistic regression analysis of factors affecting diagnostic success.

	Diagnosed (n = 69)	Undiagnosed (n = 46)	Univariable Analyses		Multivariable Analyses	
			OR (95% CI)	*p*-Value	OR (95% CI)	*p*-Value
Age, years	65.7 ± 10.2	66.2 ± 10.7	1.00 (0.97–1.04)	0.828		
Male sex	24 (34.8)	16 (34.8)	1.00 (0.46–2.19)	1.000		
Characteristics						
Mixed GGO	65 (94.2)	39 (84.8)	2.92 (0.80–10.61)	0.113		
Pure GGO	4 (5.8)	7 (15.2)	1.00			
Size (mm), long axis	21.9 ± 7.3	17.1 ± 6.6	1.11 (1.04–1.18)	0.001	1.10 (1.00–1.16)	0.042
Bronchus sign in CT						
Positive	55 (79.7)	18 (39.1)	6.11 (2.66–14.07)	<0.001		
Negative	14 (20.3)	28 (60.9)	1.00			
EBUS image						
Mixed blizzard	56 (81.2)	7 (15.2)	24.00 (8.78–65.61)	<0.001	20.92 (7.50–58.31)	<0.001
Others	13 (18.8)	39 (84.8)	1.00			

Values are presented as mean ± standard deviation or number (%). CT: computed tomography; EBUS: endobronchial ultrasound; GGO: ground-glass opacity.

## Data Availability

The data presented in this study are available on request from the corresponding author.

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
