# Peer review of "Endobronchial Ultrasound Using Guide Sheath-Guided Transbronchial Lung Biopsy in Ground-Glass Opacity Pulmonary Lesions without Fluoroscopic Guidance"

_cancers, 2024, doi:10.3390/cancers16061203_

Round 1

Reviewer 1 Report

Comments and Suggestions for Authors

This study demonstrates the utility of EBUS-GS without fluoroscopy for GGO lesions.

Major comment

1.     In the first place, it is not clear why this research was conducted without using X-ray fluoroscopy. Does this facility perform all EBUS-GS procedures without using X-ray fluoroscopy? Previous reports have shown that the diagnosis rate improves when lesions can be confirmed with X-ray fluoroscopy, and not using fluoroscopy is considered to be disadvantageous.

2.     The authors should describe how they performed the bronchoscopy. The model number and outer diameter of the bronchoscope, EBUS probe, guide sheath, biopsy forceps, brush, etc. used should be listed.

3.     In order to perform a biopsy using the EBUS-GS method without using fluoroscopy, a method is needed to determine how far into the forceps to insert the biopsy. The authors should also describe the specific biopsy method in the method section.

4.     The authors should state the number of biopsies taken.

Minor comment

The authors should also add the method by which they determined the final diagnosis to the Method section. The minimum follow-up period used to determine the diagnosis of benign disease must also be stated.

Author Response

Thank you very much for your kind e-mail, which informed us that our manuscript would have a chance for a revision. We have tried to amend and improve the paper according to the reviewers’ comments. The point-by-point response to each comment suggested by the reviewers is given on separate pages. We hope that the revised version will fulfill the requirements for publication in Cancers.

We are looking forward to hearing good news from you.

Sincerely yours

June Hong Ahn, Division of Pulmonology and Allergy, Department of Internal Medicine, Yeungnam University Medical Center, College of Medicine, Yeungnam University, 170 Hyeonchung-ro, Namgu, Daegu, 42415, Republic of Korea. Tel: 82-53-640-6577, Fax: 82-53-620-3849, E-mail: fireajh@gmail.com

Reviewer 2 Report

Comments and Suggestions for Authors

This is a well written clinical study.  

I only have two small suggestions.

1)  In the undiagnosed group, presumably patients would have had resection surgery for diagnosis or close followup.  Could some information be added to the clinical outcome (ie) was there ever a diagnosis in this group?

2) Can some information on smoking status or other risk factors please be included.  Perhaps there is a relationship between say smoking status and RP-EBUS diagnosis which may help guide clinical practice.   

Author Response

(The authors gave the same response as above.)
